# Mediterranean Diet and Neurodegenerative Diseases: The Neglected Role of Nutrition in the Modulation of the Endocannabinoid System

**DOI:** 10.3390/biom11060790

**Published:** 2021-05-24

**Authors:** Federica Armeli, Alessio Bonucci, Elisa Maggi, Alessandro Pinto, Rita Businaro

**Affiliations:** 1Department of Medico-Surgical Sciences and Biotechnologies, Sapienza University of Rome, Corso della Repubblica, 79, 04100 Latina, Italy; Federica.armeli@uniroma1.it (F.A.); bonucci.alessio@libero.it (A.B.); elisa.maggi@uniroma1.it (E.M.); 2Department of Experimental Medicine, Sapienza University of Rome, 00161 Rome, Italy; alessandro.pinto@uniroma1.it

**Keywords:** endocannabinoids, mediterranean diet, neuroinflammation

## Abstract

Neurodegenerative disorders are a widespread cause of morbidity and mortality worldwide, characterized by neuroinflammation, oxidative stress and neuronal depletion. The broad-spectrum neuroprotective activity of the Mediterranean diet is widely documented, but it is not yet known whether its nutritional and caloric balance can induce a modulation of the endocannabinoid system. In recent decades, many studies have shown how endocannabinoid tone enhancement may be a promising new therapeutic strategy to counteract the main hallmarks of neurodegeneration. From a phylogenetic point of view, the human co-evolution between the endocannabinoid system and dietary habits could play a key role in the pro-homeostatic activity of the Mediterranean lifestyle: this adaptive balance among our ancestors has been compromised by the modern Western diet, resulting in a “clinical endocannabinoid deficiency syndrome”. This review aims to evaluate the evidence accumulated in the literature on the neuroprotective, immunomodulatory and antioxidant properties of the Mediterranean diet related to the modulation of the endocannabinoid system, suggesting new prospects for research and clinical interventions against neurodegenerative diseases in light of a nutraceutical paradigm.

## 1. Endocannabinoid System: Physiology and Pathophysiology No

Initially analyzed for its ability to modulate pain, the endocannabinoid system is now considered a powerful system, capable of regulating several physiological and pathological processes. Interest in endocannabinoid activities has increased considerably in recent years, since their involvement in the fine-tuning of numerous metabolic processes, to maintain cell and tissue homeostasis, has been clearly established [1]. Endocannabinoids (ECs) exhibit pleiotropic activities impacting cognition, pain, appetite and inflammatory processes. They are involved in glucose and lipid metabolism, and they stimulate food uptake, so their chronic overactivation leads to the development of obesity, insulin and leptin resistance, metabolic syndrome and type 2 diabetes [2].

Initially, interest in this class of compounds was aroused by the observation that the endogenous substances under study were able to bind to the specific receptors for some psychoactive molecules obtained from *Cannabis sativa*, and for this reason, one of the first endocannabinoids to be identified was named anandamide from a Sanskrit term that means happiness. Since then, the role of endocannabinoids in the control of different metabolic pathways to maintain homeostasis has emerged.

The EC system constitutes a complex endogenous system of communication between cells. As detailed below, the system of ECs is made up of three groups of elements that work in synergy and are stimulated as needed, depending on the contingent onset of an imbalance. Therefore, the levels of ECs vary on demand, according to homeostatic needs, making it difficult to quantify a normal range of the single components of the system [3]. Moreover, given the complexity of the system, the heterogeneity of the molecules that are part of it, and the variety of biological mechanisms underlying their action, research in this area has to overcome many problems. Results obtained so far have established that several molecules are involved in this system: receptors, ligands and enzymes responsible for EC synthesis and degradation [4]. The biological activity of endocannabinoids (ECs) is primarily mediated by two receptors: cannabinoid receptor 1 (CB1R) and cannabinoid receptor 2 (CB2R), belonging to the superfamily of G protein-coupled receptors. CB1Rs, the most abundant in the central nervous system (CNS), are located mostly on the neuronal terminals of brain regions responsible for motor coordination, such as the cerebellum, striatum and substantia nigra, and within the prefrontal cortex, hippocampus, amygdala and hypothalamus where they control memory and cognitive functions, fertility, sexual behavior and food intake [5,6]. ECs regulate appetite and food intake via activation of CB1R localized within brain regions controlling feeding, energy expenditure and reward, such as the hypothalamus and limbic forebrain, suggesting their involvement in both the homeostatic and hedonic control of eating [7,8].

CB1Rs are also expressed, in lower amounts, in cells of peripheral organs and tissues, including hepatocytes, adipocytes, gastrointestinal tract and muscle tissue, where CB1Rs modulate metabolism [6]. On the other hand, CB2Rs play a crucial role in the CNS immune response: indeed, their modulation impacts the migration and release of inflammatory mediators by activated microglia and astrocytes. They are also expressed in immunocompetent cells such as monocytes, T lymphocytes and mast cells, influencing their activity [9,10,11].

CB2Rs are additionally expressed on enteric neurons and epithelial cells of the digestive tract [12].

CBR-related Gi/O proteins inhibit adenylate cyclase (AC) activity, decreasing intracellular levels of second messenger cyclic adenosine monophosphate (cAMP), modulating excitatory or inhibitory neurotransmission through inhibition of Ca^2+^ entry. CB1Rs activate various kinases too, particularly mitogen-activated protein kinases (MAPK) [10,13]. They affect excitotoxicity and cellular survival and counteract inflammatory processes [9]. There are other receptors engaged in endocannabinoid signaling: transient receptor potential vanilloid type 1 (TRPV1), relevant to synaptic plasticity; and peroxisome proliferator-activated receptor-α (PPAR-α) and peroxisome proliferator-activated receptor-γ (PPAR-γ), involved in neuroprotection and energy balance modulation, lipid metabolism and adipogenesis [14,15]. A 2018 clinical trial revealed that PPAR-α was implicated in appetite control in overweight patients [16].

Our knowledge about the mechanisms underlying CBR function derives from studies dealing with natural and synthetic cannabinoid compounds, including Δ 9-tetrahydrocannabinol (Δ9-THC) and cannabidiol (CBD), the major components of the *Cannabis sativa* plant [14]. The most characterized ECs are anandamide (AEA) and 2-arachidonoylglycerol (2-AG), lipid molecules that originate from arachidonic acid. AEA is synthesized by N-acetyl phosphatidylethanolamine phospholipase D (NAPE-PLD), while 2-AG secretion depends mostly on the activity of phospholipase C, diacylglycerol lipase α (DAGLα) and diacylglycerol lipase β (DAGLβ) [17]. The ECs are hydrolyzed by two main enzymes: fatty acid amide hydrolase (FAAH) and monoacylglycerol lipase (MAGL), targeting AEA and 2-AG, respectively [18]. A bulk of studies have examined the role of ECs in neurodegenerative diseases (NDDs) such as Parkinson’s disease (PD), Alzheimer’s disease (AD) and multiple sclerosis (MS), where inflammation plays a critical role and the endocannabinoid system has been shown to be altered both in experimental models and in patients [14,19,20,21]. Modulation of the endocannabinoid system was detected during AD progression, characterized by the accumulation of amyloid β peptide (Aβ), which induces persisting glial activation and neuroinflammation [22]. In AD transgenic mice, which overexpress a mutant form of the amyloid precursor protein, a depletion of AEA was detected within the hippocampal area [23]. Preclinical studies in murine models have demonstrated AD-like symptom improvement after FAAH and MAGL inhibition [24,25]. In AD patients, an increased FAAH expression was detected in peripheral blood mononuclear cells [26]. APP/PS1 mice, expressing the mutant form of amyloid precursor protein and presenilin, treated with CB2R agonists, showed improved cognitive performance, paralleled by a lower microglial activity, suggesting that the neuroprotective role of CB2R receptors was dependent on reductions in both neurotoxicity and neuroinflammation, as well as on increased Aβ clearance, leading to improved memory and cognition [23,27]. In support of this result, a follow-up study, performed in knockout CB2R mouse models of AD, showed increased levels of Aβ [28]. Dysregulation of the endocannabinoid system has also been evidenced in animal models of Parkinson’s disease (PD) [20]. CB2R activation, in mice treated with 6-hydroxydopamine (6-OHDA) and 1-methyl-4-phenyl-1, 2, 3, 6-tetrahydropyridine (MPTP), reduces depletion of dopamine and neuroinflammation [29,30,31]. In rodents treated with lipopolysaccharide (LPS) at the level of the nigrostriatal area to simulate parkinsonian inflammation, CB2R activation was able to downregulate the expression of inflammatory mediators [32]. In MPTP-treated mice, MAGL inhibition combined with CB2R activity has a neuroprotective effect, while FAAH inhibition by URB597 showed antiapoptotic and anti-inflammatory effects accompanied by improved motor behavior [33,34,35]. Dysregulation of EC levels has been demonstrated both in PD rat models, at the level of the globus pallidus and substantia nigra, and in cerebrospinal fluid of PD patients [36,37]. Neuroimaging demonstrated a different expression of CBRs in PD patients’ substantia nigra compared with dopaminergic projection regions [38]. Changes in the endocannabinoid system have also been highlighted in multiple sclerosis (MS). Different studies in MS mouse models have shown positive effects following the activation of CBRs [14]. In murine Theiler’s encephalitis virus-induced demyelinating disease (TMEV-IDD) and chronic relapsing autoimmune encephalomyelitis (CREAE), treatment with CBR agonists revealed a beneficial impact on inflammation, improving tremor and spasticity [39,40]. Immunohistochemical analysis in post-mortem brains of MS patients showed an upregulation of microglia CB2Rs and increased amounts of FAAH metabolites [41]. In both in vitro and in vivo investigations, phytocannabinoids (THC and CBD) exhibited immunoregulatory properties in chronic neuroinflammation, improving symptoms and quality of life for NDD patients [14]. Furthermore, lipid compound members of the same family of canonical ECs, such as palmitoylethanolamide (PEA) and oleoylethanolamide (OEA), N-acetylanolamines hydrolyzed by N-acylethanolamine acid amidase (NAAA), show anti-inflammatory properties [42,43,44,45]. PEA and OEA are endogenous PPAR agonists; they regulate the release of pro-inflammatory mediators, food intake, weight loss, lipolysis, and also indirectly activate CB1Rs and CB2Rs [42,46]. In a recent study on different cell lines (SH-SY5Y, C6, BV2), increased PEA levels, subsequent to downregulation of NAAA, counteracted LPS- and interferon-γ (INF-γ)-induced inflammation [47].

Taking together these observations, modulation of the endocannabinoid system could be a potential wide-ranging therapeutic target in reducing the main hallmarks of neurodegeneration, including neuroinflammation, excitotoxicity and oxidative stress (OxS) [44]. There are well-established neuroprotective properties of the Mediterranean lifestyle, but it is not yet known whether its nutritional and caloric balance can induce a modulation of the endocannabinoid system. This review aims to reconsider some neuroprotective properties of a typical Mediterranean diet (MedD) as being linked to a rebalancing between the components of the endocannabinoid system.

## 2. Mediterranean Diet

It is estimated that the prevalence of NDDs in elderly people, including AD and PD, will increase in the next few decades [48]. Beyond the development of new specific drugs to restrain the neurological symptomatology, it is necessary to individuate new prevention potential strategies. In this scenario, a healthy lifestyle, which includes an adequate dietary pattern and physical activity, seems to have a pivotal role [48,49]. A chronic low-grade inflammatory status, which induces microglial chronic activation leading to pro-inflammatory cytokine production and neuronal apoptosis, is recognized as a key factor in the pathogenesis of AD and other NDDs [50]. The Whitehall II prospective cohort study (2017) showed how a pro-inflammatory Western diet, which increases serum interleukin-6 (IL-6) levels, may accelerate cognitive dysfunction [51]. Evidence supports the role of the Mediterranean diet (MedD) in the primary and secondary prevention of non-communicable chronic diseases (NCDs) such as cardiovascular disease, AD and dementia [52,53,54,55]. The high intake of different types of fruit, seasonal green leafy vegetables, extra-virgin olive oil (EVOO) (cold pressed), fresh blue fish, whole grains, legumes, nuts, spices and a low intake of alcohol such as red wine, with small quantities of red meat, eggs and dairy products, defines the Mediterranean dietary pattern [48,56,57]. The most recent revision of the Mediterranean Diet Pyramid (MDP), in addition to graphically summarizing all the nutritional aspects, also emphasizes sociocultural, environmental and sustainability issues, which are essential components of the concept of well-being >(Figure 1) [58].

Daily intake of whole grains, fruits and vegetables containing phenolic compounds such as flavonoids and polyphenols, and sources of vitamins, fiber, potassium, magnesium and folic acid, has been shown to reduce human body fatness and the risk of death from stroke, cardiovascular disease, diabetes and cancer [56]. Phytosterols, dietary fiber, and flavones from legumes, following microbiota metabolic transformation and intestinal absorption, produce beneficial effects on cholesterol metabolism and regulate blood pressure and insulin sensitivity, leading to a decrease in mortality risk [56,57,59]. The intake of fish and nuts, which are rich in polyunsaturated fatty acids (PUFA n-3, n-6), such as alpha-linolenic acid, EPA (eicosapentaenoic acid), DHA (docosahexaenoic acid) and linoleic acid, provides lipoprotein metabolism benefits [56,60]. EVOO is the main or exclusive fatty seasoning in the MedD, a valuable source of monounsaturated fatty acids (MUFAs) and phytosterols which modulate the lipid metabolism, as well as of polyphenols with antioxidant and anti-inflammatory activity [56].

Some scientific studies have raised the issue of the lectins present in some foods of the Mediterranean diet (especially whole grains and beans) with respect to the alteration of intestinal permeability and the consequent pro-inflammatory response [61,62]. Petroski W. and Minich D.M. (2020), more recently, published a narrative review, which concluded that the available human trials do not provide strong evidence to claim that lectin-rich cooked foods increasing intestinal permeability cause inflammation in the general population, and which highlighted the limitations of the studies conducted in animal models and cell cultures using isolated lectins. This scenario is quite far from the reality, in which the lectin intake is relatively small and is combined with other bioactive components [63,64]. However, several studies support the role of the Mediterranean diet in reducing the risk of non-communicable disease through various mechanisms, including modulation of the inflammatory response [65,66,67,68], which has also been related to the effects of the Mediterranean diet on intestinal permeability and microbiota, although further studies are required to reach definitive conclusions in relation to the latter hypothesis [69].

The MedD, through molecules such as β-carotene, phenolic components, vitamin C and vitamin E, enhances the activity of the biological antioxidant systems. In two randomized and controlled trials including overweight and healthy subjects, the intake of a diet rich in EVOO and polyphenols was associated with a significant reduction in 8-oxo-deoxyguanosine and 8-isoprostane in urinary samples [70,71]. Antioxidant compounds and/or compounds modulating the human antioxidant system activity introduced with a MedD, such as flavonoids and polyphenols, lower neuro-inflammatory events by blocking the activity of transcription factors, such as nuclear factor-kappa B (NF-kB), and the expression of pro-inflammatory cytokines. Several biological activities of these foods’ compounds are mediated by the activation of PPAR-γ, which promotes the release of anti-inflammatory cytokines and inhibits NF-κB [56]. The intake of a diet enriched with EVOO decreased the expression of some genes linked to pro-inflammatory pathways, including C-C motif chemokine ligand 3 (CCL3), C-X-C motif chemokine ligand 1/2/3 (CXCL1/CXCL2/CXCL3), C-X-C motif chemokine receptor 4 (CXCR4), interleukin-1β (IL-1β), IL-6 and oncostatin M (OSM), in peripheral blood mononuclear cells of individuals with metabolic syndrome, if compared with diets with low phenolic compound content [72]. In vitro, inhibition of NF-kB by resveratrol (RSV), a polyphenol found in red wine, grapes and peanuts, supported its anti-inflammatory activity [73,74]. Moreover, the PUFA n-3, found in fish oil, suppresses systemic inflammation and neuroinflammation, whereas saturated fatty acid (SFA) promotes inflammation. In addition, diet impacts the intestinal microbiota composition: a healthy diet favors the growth of beneficial symbiotic bacteria and counteracts the growth of pathogens. Microbiota metabolic products may affect gut permeability and stimulate both peripheral and CNS inflammation [50]. A very recent study dealing with a cohort of Southern Italian patients with MS demonstrated that adherence to the MedD positively affected several neurodegenerative parameters, due to modulation of the gut microbiota and systemic inflammation [75]. The brain inflammation is significantly reduced in relationship to a healthy lifestyle with nutritional habits based on antioxidant and anti-inflammatory components, balancing the intestinal microbiome, which affects the brain innate immune cells through the systemic circulation [50]. A diet enriched with EVOO supplemented to TgSwDI mice induced a significant decrease in brain Aβ and neurofibrillary tangles and an improvement in cognitive performance. Consumption of a diet with EVOO following AD onset had minor effects, but still increased Aβ clearance through the blood–brain barrier (BBB) [76]. In AD, Aβ modifies glucose metabolism in neural cells and thus insulin resistance has a relevant impact on the progression of this pathology: indeed, diabetes mellitus is often accompanied by the development of NDDs. In an in vitro study, hydroxytyrosol, a polyphenol present in EVOO, added to an astrocytic cellular line in the presence of Aβ, reduced its cytotoxic effect, by the activation of protein kinase B (Akt) and inhibition of mammalian target of rapamycin (mTOR), reverting the insulin signaling pathway, showing that olive oil consumption prevents cognitive decline by improving insulin metabolism [77]. In vitro studies on PC12 cells supported the beneficial effect of RSV on Aβ-induced neurotoxicity [78].

Several clinical trials support the hypothesis that the MedD exerts its neuroprotective action against cognitive decline by its antioxidant and anti-inflammatory properties [79]. In a long-term primary prevention clinical trial, Prevención con Dieta Mediterránea (PREDIMED), a comparison was made between groups of older adults with elevated vascular risk: one group was consuming a low-fat diet, enriched with nuts and olive oil, and a second group was consuming a low-fat diet; a higher score in cognitive performance was obtained by the first group [80]. A further randomized clinical trial conducted in elderly subjects at high vascular risk, cognitively healthy, showed that the MedD, rich in antioxidants and supplemented with EVOO or nuts, had a beneficial effect on cognitive functions compared with a Western diet [81]. In a recent randomized clinical trial, PD patients, who adhered to a Mediterranean personalized diet for 10 weeks, obtained higher scores on the cognitive evaluation compared with control groups [82]. Another clinical study revealed that a vegetable-based nutritional protocol, with a moderate consumption of meat and alcohol, can prevent PD [82]. Two trials in Japan showed that a healthy diet, rich in fruits and vegetables, confers protection against development of PD and ALS [83,84]. MedD reduced motor impairment and improved quality of life when supplemented to Huntington’s disease (HD) patients [85].

Limited and conflicting evidence on the MedD’s influence on NDDs, such as PD, ALS and HD, necessitates further investigations, especially on the synergistic effect of multiple foods [48]. Finally, the main feature of the MedD is the high plant/animal food ratio resulting in a particular richness of biological compounds that interact with the endocannabinoid system. Few human studies have systematically explored this therapeutic possibility. Furthermore, personalized dietary protocols aiming at improving all NDDs by balancing endocannabinoid tone have not yet been set up. As described below, the evidence obtained so far from both in vitro and in vivo studies suggests treating NDDs by nutritional protocols based on a Mediterranean dietary pattern, affecting the endocannabinoid system (Figure 2).

## 3. CB2R Ligands: β-Caryophyllene and 3,3′-Diindolylmethane

β-Caryophyllene (BCP) is a bicyclic sesquiterpene widely distributed in the plant kingdom and the most extracted terpenoid from *C. sativa* [86,87]. BCP is present in numerous spices and foods, including black pepper (*Piper nigrum*) (7.29%), hops (*Humulus lupulus*) (5.1–14.5%), oregano (*Origanum vulgare*) (4.9–15.7%), cloves (*Syzygium aromaticum*) (1.7–19.5%), valerian (*Valeriana officinalis*), wild sage (*Salvia verbenaca*), rosemary (*Rosmarinus officinalis*) (0.1–8.3%) and basil (*Ocimum basilicum*), as an essential oil [88]. Depending on the forage quality, BCP can also accumulate in cow’s milk in higher or lower concentrations [89]. The safety of the molecule has been approved by the Food and Drug Administration and the European Food Safety Authority, who have approved its use as an additive in cosmetics and foodstuff production [90]. Previous works established its antibacterial, antifungal, anticancer, acetylcholinesterase-inhibitory, anti-inflammatory and antioxidant properties, which made it a promising candidate for the treatment of many acute and chronic diseases [91]. At 100 nM, BCP acts as a selective full agonist of CB2Rs [92]. Thanks to its high liposolubility, BCP can cross the BBB, exerting broad-spectrum neuroprotective activity against many NDDs, such as PD, MS and AD [93] (Figure 3).

Neuroinflammation, perpetrated by M1 microglial cell chronic activation, is an important driver mechanism in neuropathology [94]. In an amyloid β1-42 (Aβ1-42)-induced neuroinflammation model, BCP (10, 25 and 50 μM) treatment attenuated the release of nitric oxide (NO), prostaglandin E2 (PGE2) and pro-inflammatory cytokines in BV2 microglial cells, controlling toll-like receptor 4 (TLR4) overexpression and IκBα/NF-κB downstream activity [95]. Likewise, in transgenic APP/PS1 mice, oral administration of BCP (16, 48 and 144 mg/kg) improved neurobehavioral hallmarks of an Alzheimer-like phenotype, reducing Aβ burden, astrogliosis and levels of the pro-inflammatory mediators (cyclooxygenase 2 (COX-2), IL-1β and tumor necrosis factor-α (TNFα)) in the cerebral cortex and hippocampus [96]. In a Wistar rat model of PD, previous administration for 4 weeks of BCP (50 mg/kg) attenuated rotenone-induced toxicity in brainstem dopaminergic neurons and controlled neuroinflammation, decreasing inducible nitric oxide synthase (iNOS), COX-2, TNFα, IL-6 and IL-1β expression [97]. BCP treatment has been shown to improve autoimmune encephalomyelitis (EAE), in a mouse model of MS, preventing axonal demyelination through the modulation of Th1/Treg immune balance and the mitigation of microglial cells, and CD4+ and CD8+ T lymphocyte activity [98]. Although the exact mechanism by which BCP exerts its anti-inflammatory activity (already reached at 4 mg/kg/day) is still not known, there is a common agreement that CB2R represents a key player in these processes [99,100]. Misfolding proteins, OxS and many pro-inflammatory cytokines trigger the TLR4/p38 mitogen-activated protein kinase (p38 MAPK) pathway, which in turn leads to nuclear translocation of nucleus of activator protein 1 (AP-1) and NF-κB transcription factors [101]. AP-1 and NF-κB upregulate pro-inflammatory mediators, promoting M1 microglial shift [102]. CB2R-dependent inhibition of p38 MAPK/NF-κB signaling has been suggested as a possible mechanism involved in BCP anti-inflammatory effects [103]. Although CBRs are Gi-coupled receptors, CB2Rs enhance cAMP synthesis and the cAMP-dependent protein kinase (PKA)/cAMP-response element binding protein (CREB) downstream pathway. CREB promotes TGF β, IL-4, IL-10 and CD206 expression, typical markers of an M2 anti-inflammatory microglial phenotype [104]. In a similar fashion, BCP may induce M2 microglial shift via PPAR-γ [105]. In a randomized pilot clinical trial lasting 18 months, pioglitazone (a PPAR-γ agonist) in nondiabetic patients with AD significantly improved memory and cognitive performances [106]. It is well-established that PPAR-γ’s functions are regulated by its main coactivator, peroxisome proliferator-activated receptor γ coactivator-1α (PGC-1α) [107]. Additionally, the expression of PGC-1α can be increased by Sirtuin-1 (Sirt1), a type 3 histone deacetylase, which is assisted by CREB phosphorylation. Taking together these observations, BCP could lead to a M1/M2 microglial shift via CB2R/cAMP/PKA/CREB/Sirt1 signaling, upregulating PGC-1α and the further activation of PPAR-γ [108,109]. OxS is typically involved in neuronal damage and NDD progression [110]. In an in vitro study on the C6 glioma cell line, BCP at a concentration of 0.5 or 1 μM counteracted glutamate-induced OxS cytotoxicity, restoring the glutathione (GSH) antioxidant system and mitochondrial membrane potential [111]. In a similar fashion, BCP (10, 25 and 50 μM) inhibited NO production in BV2 microglial cells [95]. In animal models of NDDs, BCP at doses of 25 and 50 mg/kg/day has been found to control NO and hydrogen peroxide (H_2_O_2_) production in EAE mice [112]. The administration for 4 weeks of BCP (50 mg/kg) proved effective in reducing lipid peroxidation, implementing the reserves of GSH and antioxidant enzymes, such as superoxide dismutase (SOD) and catalase (CAT), in PD-like Wistar rats [97,113]. Although part of BCP scavenger activity is directly due to its cyclic structure, CB2R downstream signaling also seems to be important against neuronal OxS [111,114]. Interestingly, Assis et al. (2014) found that BCP’s cytoprotective effects on the C6 glioma cell line were mediated by nuclear factor erythroid 2-related factor 2 (Nrf2) in a CB2R-mediated manner [111]. Nrf2, a member of the cap “n” collar (Cnc) family of transcription factors, is the main cellular sensor of oxidoreductive homeostasis [108,115]. Under OxS conditions, the presence of reactive oxygen species (ROS) and peroxidized lipids, such as 4-hydroxy-2-nonenal (4-HNE), trigger Nrf2 nuclear translocation. Nrf2 binding to antioxidant responsive element (ARE) promotes the transcription of detoxifying enzymes, including superoxide dismutase (SOD), heme oxygenase-1 (HO-1), CAT, glutamate cysteine ligase (CGL), glutathione reductase (GR), peroxiredoxin (Prx) and thioredoxin (Trx), and/or proteins with thiol (-SH) group such as GSH, implicated in cytoplasmic and mitochondrial antioxidant endogenous defense restoration [108,116]. Several works have demonstrated the presence of CB2Rs at the level of the prefrontal cortex, hippocampus, basal ganglia and cerebellum, whose modulation may be a promising therapeutic target in AD, PD, HD and hereditary spinocerebellar ataxias in order to counteract the pathologic accumulation of free radicals [97,117,118,119].

The question of which molecular cascade links CB2R to Nrf2 still remains elusive; however, Sirt1 has been found to upregulate Nrf2 expression [120]. It is possible that the cAMP/PKA/CREB/Sirt1 pathway may play an important role, but more studies will be needed to confirm this hypothesis. Moreover, Sirt1 itself: (i) implements the efficiency of the mitochondrial electron transport chain (ETC), stimulating the expression of uncoupling proteins (UCPs) 2, 4 and 5 [121]; (ii) synergistically with Nrf2, upregulates PCG-1α expression which represents a key step in counteracting neuronal mitochondrial dysfunction, promoting mitochondrial biogenesis [122,123]; and (iii) Nrf2, like Sirt1, promotes M2 microglial shift, reducing OxS as well as related neuroinflammation [124]. On the other hand, a recent study demonstrated that in microglia cells, the CNR2 gene (CB2R gene) owns an ARE sequence in the promoter, harnessing a positive feedback loop between Nrf2 activation and upregulation of CB2R transcription which may counter the pathological positive feedback loop between TNFα and NF-κB activation [125,126]. A third neuroprotective activity of BCP is the induction of brain-derived neurotrophic factor (BDNF) synthesis in a CB2R-dependent manner [127]. BDNF is a neurotrophin, which plays a prominent role in neuronal survival, plasticity and repair as well as in mood and cognitive function regulation [128]. In many NDDs and chronic inflammatory disorders, BDNF levels are reduced [129]. BCP, triggering the CB2R/cAMP/PKA/CREB pathway, enhances PCG-1α levels, which in turn stimulates neuronal synthesis of FNDC5, a myokine secreted by muscle and brain, which finally leads to BDNF expression [128]. Furthermore, high levels of BDNF increase CREB phosphorylation and PCG-1α expression, amplifying BDNF transcription in a positive feedback loop [130]. In microglial cells, BCP upregulates downstream M2-associated mediators, including BDNF, through a CB2R/Nrf2 signaling cascade [124]. It has also been reported that neural progenitor cells express CB2R, which seems to be important for their proliferation and differentiation, opening a promising line of research [131,132] (Figure 3).

The maintenance of neurovascular coupling, resulting from a complex interaction between brain microvascular endothelial cells, astrocytes and neurons, is under the precise control of CB2Rs [133]. Maintenance of BBB integrity is mandatory for brain homeostasis, as it inhibits leukocyte infiltration and promotes Aβ clearance [9]. Tian et al. (2016) performed an in vitro study on co-cultures of brain microvascular endothelial cells, neurons and astrocytes to investigate the effects of BCP on oxygen-glucose deprivation and re-oxygenation-induced injury: BCP (10 μM/L) downregulated Bax and matrix metallopeptidase 9 (MMP-9) expression, and upregulated claudin-5 (CLDN-5), zonula occludens-1 (ZO-1), occludin, Bcl-2, and growth associated protein 43 (GAP-43) expression, reducing BBB permeability [134]. It has been suggested that this pro-homeostatic effect of BCP on BBB also involves both the AMPK/cAMP and Nrf2/HO-1 pathways in a CB2R-dependent [127,135] (Figure 3).

Glucosinolates (GLs) belong to a structurally homogeneous class of thiosaccharidic metabolites, mainly contained in roots, seeds and leaves of cruciferous vegetables (Brassicaceae family) such as broccoli (*Brassica oleracea* var. italica), cabbage (*B. oleracea* var. capitata f. alba), cauliflower (*B. oleracea* var. botrytis) and horseradish (*Armoracia rusticana*) [136,137]. The anticarcinogenic action of GLs has been known for decades, as well as their antioxidant and immunomodulatory properties: regular consumption of cruciferous vegetables is associated with a reduced incidence of cancer [138,139]. Brassicaceae vegetables contain 206–3895 mg/kg of GLs; however, most of these are affected by the cooking processes, so the greatest health benefits seem to be related to their raw intake [140]. A regular dietary consumption of cruciferous vegetables has also been linked to several benefits in NDDs [141]. A specific GL, 3,3′-diindolylmethane (DIM), might partly mediate its neuroprotective actions by enhancing endocannabinoid tone. Specifically, DIM is an indole-3-carbinol dimer, noted for its hepatoprotective, antioxidant and anticancer properties [142,143,144]. It is a selective aryl hydrocarbon receptor modulator, which acts as an agonist or antagonist in a tissue-specific manner, both in the CNS and in peripheral tissues [145]. Investigating DIM pharmacokinetics, Anderton et al. (2004) observed that it reaches a concentration of 5–36.5 μM in the brain within 6 h after oral administration to mice (250 mg/kg) [146]. Indirectly, this work also demonstrated that the molecule can cross the BBB. A clinical trial on healthy subjects reported that up to a single dose of 300 mg, DIM does not induce relevant side effects, although no difference in Cmax emerged between the 200 mg dose when compared with that of 300 mg [147]. The tolerability of the molecule was also confirmed in a pilot study on postmenopausal women with a history of early-stage breast cancer who received an administration of 108 mg DIM/day for 30 days [148]. Based on body surface area, DIM at a dose of 0.8–1.6 mg/kg could be effective in exerting biological activity in humans [149,150]. In vitro, DIM reduces inflammatory mediator release by macrophages and microglial cells after an inflammatory LPS stimulus. Kim et al. (2013) found that both in vitro, on BV-2 microglia, and in vivo, on C57BL/6 mice, DIM respectively downregulated iNOS and COX-2 expression, and attenuated hippocampal inflammation after LPS stimulus, inhibiting NF-κB [151,152]. The p38 MAPK signaling pathway plays an important role in triggering both inflammatory cascade and intrinsic pathway of neuronal apoptosis [153]. In a recent study on mouse hippocampal cell cultures treated with DIM (0.01, 0.1, 1, 10 μM), the latter was shown to exert neuroprotective effects against ischemia-induced cell damage, decreasing the levels of pro-apoptotic factors such as caspase-3, ischemia-induced lactate dehydrogenase and p38 MAPK [154]. In a similar fashion, DIM and its analog (DIM-CpPhtBu) were found to be effective in inhibiting apoptosis in vitro and to exert a cytoprotective effect in in vivo models of PD [155,156]. Interestingly, DIM-C-pPhtBu seems to induce neuroprotection, acting as a PPAR-γ agonist [157]. DIM, both in vitro and in vivo, increases the expression of BDNF and antioxidant enzymes (HO-1, CGL and NAD(P)H quinine oxidoreductase-1), significantly reducing hippocampal neuronal depletion [150]. It is possible that DIM exerts neuroprotection through two molecular cascades: the tropomyosin-related kinase receptor B (TrkB)/Akt/CREB/BDNF pathway [158], and the Nrf2/ARE or TrkB/Akt/Nrf2/ARE pathway [159]. In addition, TrkB and BDNF establish a positive feedback loop, reversing scopolamine-induced memory impairment (DIM 10–20 mg/kg) [150]. Taking together these overlaps with BCP downstream biological mediators (e.g., p38MAPK, Nrf2, PPAR-γ, CREB, etc.), a possible role for CB2Rs in the neuroprotective activity of DIM has been proposed [160]. To our knowledge, only one study found that DIM can act as a CB2R partial agonist, with a binding affinity of around 1 μM [161]. If this finding were to be confirmed by further studies, it would provide a new insight into the biological activity of DIM (Figure 3).

## 4. FAAH Inhibitors: Flavonoids

There is robust evidence that, by promoting endocannabinoid tone through FAAH inhibitors, an overall improvement in many hallmarks of neurodegeneration will take place, with promising results in terms of efficacy and tolerability [25]. Current developments of FAAH inhibitors are aimed at synthesizing compounds that, at the same time, activate receptors, such as the PPAR family, and/or inhibit COX-2 [162,163]. The discovery that AEA and 2-AG are not only substrates of FAAH but also of PPAR-γ and COX-2 suggested that there may be an overlap of structural requirements for the same molecule to act simultaneously on three targets. In support of this theory, indomethacin and ibuprofen, two non-steroidal anti-inflammatory drugs (NSAIDs), are COX inhibitors and modulators of PPAR-γ and FAAH [164,165]. On the other hand, several compounds, such as flavonoids, acting as PPAR-γ agonists are also FAAH inhibitors [166,167]. Flavonoids are a subclass of polyphenolic compounds with a 15-carbon common structure, arranged in two phenyl rings and a six-membered heterocyclic ring [168]. Based on the mutual connection between B and C rings, and the degree of hydroxylation, oxidation and saturation of the heterocyclic ring, flavonoids are classically divided into six classes: flavonols, flavones, flavan-3-ols (or catechins), flavanones, isoflavones and anthocyanidins [169]. They are ubiquitously distributed in MedD foodstuffs, especially in legumes, fruits and vegetables, noted for their remarkable anti-inflammatory, antioxidant and cytoprotective properties against several chronic diseases [169,170]. Only a few studies have evaluated the FAAH-inhibiting property of flavonoids; however, it could represent a new therapeutic prospect against NDDs. Considering that metabolism and bioavailability of flavonoids in mammals is affected by a complex network of variables including genetic polymorphisms, intestinal transit time and gut microbiota, their intake should be included in appropriate dietary protocols [171]. Biochanin A (4′-methoxy-5, 7-dihydroxy isoflavone) (BCA) is an isoflavone, mainly contained in legumes of the Fabaceae family (soy and red clover), but peanuts (*Arachis hypogaea*) and chickpeas (*Cicer arietinum*) are nevertheless other sources [172,173]. Of the overall total, BCA is the most represented flavonoid in chickpea, approximately yielding 15.7 mg/150 mg [174]. Its possible use within a nutraceutical approach to various diseases for its antioxidant, anti-inflammatory, anticancer, neuroprotective, hepatoprotective and antimicrobial properties has been considered [175]. To our knowledge, only Thors et al. (2010) reported an FAAH-inhibitory activity of BCA [176]. According to this study, BCA in vitro inhibits rat, mouse and human FAAH (IC50 respectively 1.4, 1.8 and 2.4 μM), preventing 0.5 μM AEA hydrolysis. On the other hand, in an in vivo model of persistent pain, 100 μM BCA locally injected prevented extracellular signal-regulated kinase (ERK) phosphorylation induced by intraplantar formalin injection in C57BL/6 mice. Interestingly, the latter effect was suppressed by AM251 (30 μM i.p. administration), a CB1R antagonist/inverse agonist, suggesting that this BCA activity is dependent on AEA accumulation [176]. These preliminary results suggest that BCA exerts a peripheral antinociceptive effect driven by FAAH inhibition; however, its activity in the CNS remains uncertain. Secondly, implementing its dietary intake might be insufficient to achieve a therapeutic concentration: in a group of fourteen volunteers, 80 mg of isoflavone extract from red clover was administered for two weeks but the maximum plasma concentration of BCA averaged 48 ng/mL (170 nM) [177]. However, BCA in vivo undergoes a conjugation process that finally converts it to genistein, another compound belonging to the isoflavone class, with FAAH-inhibitory activity [178,179]. In addition, BCA may exert a weak synergy with other substances, enhancing EC tone [176]. Genistein is a phytoestrogen isoflavone, produced by the metabolism of its precursors, biochanin A or formononetin, or taken up as such through diet. At micromolar concentrations, it acts as a PPAR-γ agonist and as a competitive inhibitor of FAAH (Ki value 2.8 μM) [180,181,182]. In a similar way, daidzein, a related isoflavone, activates PPAR-γ [183] and competitively inhibits FAAH (Ki value 1.7 μM) [182]. The major dietary source of genistein and daidzein in humans is soy and its derivatives. The MedD can also allow their intake, mainly through the consumption of whole grain bread and legumes such as beans (*Phaseolus vulgaris*), lentils (*Lens culinaris*), chickpeas (*Ciser arietinum*), split chicklings (*Lathyrus sativus*) and fava beans (*Vicia fava*) [184]. Despite the minor amount of genistein and daidzein in MedD foodstuffs, their potential cumulative long-term effect cannot be excluded if regularly consumed from a range of sources [185]. In a Mediterranean menu, 92% and 96% of the daily intake of genistein and daidzein, respectively, comes from brown bread (3.55 and 4.71 mg/week) [184]. The therapeutic potential of these two phytoestrogens’ supplementation in the treatment of several pathological conditions, including NDDs, is well-established because of the complex interactions between microbiota and their second metabolites; however, the exact dietary amount necessary to achieve a therapeutic effect is still not known (0.05–0.7 mg/kg for menopausal symptom treatment) [180,186,187]. Two studies on different cell cultures have demonstrated an FAAH-inhibitory property of genistein and daidzein, which was not additive for genistein to URB597 [179,182]. Whether this FAAH-inhibitory activity also takes place in vivo is still uncertain; however, it is known that the high soy dietary intake in Asian countries increases serum levels of genistein and daidzein to 2–4 μM, compatible with their Ki values [188,189].

Apigenin (4’,5,7-trihydroxyflavone) (APi) is one of the most represented flavons in the plant kingdom [190]. In recent decades, its prominent antioxidant, anti-inflammatory and neuroprotective properties have been investigated in AD and PD, both in vitro and in vivo [191]. APi interacts at micromolar concentrations with PPAR-γ, inhibits COX-2, and is a potent competitive inhibitor of FAAH [166]. Furthermore, it can also cross the BBB [192,193]. Many MedD foodstuffs contain Api, including vegetables (parsley, onions, celery), herbs (chamomile, oregano, thyme, basil), fruits (oranges) and plant-based beverages (tea, beer and wine) [194]. In a typical Mediterranean menu, Api accounts for 77% of flavone intake (6.7 mg/day), in which parsley is the primary source (96%) [184]. Based on studies in animal models, the dose required to observe a therapeutic benefit is approximately 7.5–50 mg/kg [195]. In homogenates and intact cells, Api showed a significant FAAH-inhibitory activity, making it even more attractive for NDD treatment [166]. The most well-known FAAH inhibitor/PPAR-γ activator flavonoid is kaempferol (Kmp) [196,197]. Kmp is a flavonol, widely distributed in leafy vegetables, apples, onions, broccoli, berries, tea, cabbage, broccoli, endive, kale, beans, tomato, strawberries, leek and grapes of Mediterranean countries [198]. Several molecular mechanisms have been recognized as important mediators of its neuroprotective properties, in both in vitro and in vivo models of AD and PD [198,199]. Of the twenty flavonoids tested by Thors et al. (2008), Kmp was the most potent competitive FAAH inhibitor with a Ki value of 5 μM [166]. The amount of Kmp usually consumed with a Mediterranean menu is about 7.94 mg/week, which may not be sufficient to achieve the necessary tissue concentration in vivo [184,200]. For the first time, in a contextual fear-conditioning animal model, it was demonstrated that Kmp facilitates the extinction of aversive memories, along with a reduction in anxiety in rats. The maximum therapeutic response was achieved with a 40 mg/kg dose (IC50 1 μM), while it was completely abolished by the co-administration of rimonabant, a CB1R antagonist [201].

## 5. Dietary ω-3 and ω-6 Fatty Acids Balance and Endocannabinoid System Coupling

PUFAs are key components of phospholipid membranes, as well as precursors of a large repertoire of bioactive lipid mediators. Traditionally, PUFAs are classified according to the position of the first desaturation from the methyl, n or ω terminal. The lack of expression of Δ12- and Δ15-desaturases in mammals, capable of inserting a double bond on the ω-3 and ω-6 carbons, makes α-linolenic (ALA, ω-3) and linoleic (LA, ω-6) essential fatty acids [202]. ALA and LA represent the precursors of the two most represented PUFAs in the brain, docosahexaenoic acid (DHA, ω-3 PUFA) and arachidonic acid (ARA, ω-6 PUFA), respectively [203]. Because of the low efficiency of endogenous hepatic and brain biosynthesis of DHA and ARA [204], the most efficient way to enrich tissues of these PUFAs is their direct intake from the diet [205]. In addition, N-docosahexaenoylethanolamine (DHEA) and N-eicosapentaenoyl-ethanolamine (EPEA) are two pro-homeostatic lipid mediators, obtained from subsequent conversion of DHA and eicosapentaenoic acid (EPA, ω-3 PUFA), respectively [206]. On the other hand, ARA is the precursor of endogenous ligands of CBRs, including AEA and 2-AG [207]. Previous studies found that the main source of precursors for EC synthesis in mammals derives from dietary fatty acids and PUFAs. Changes in dietary style modulate their tissue levels [208]. In this regard, the maintenance of a correct balance between ω-3 and ω-6 PUFAs plays a crucial role: for example, the usual Western diet has a ω-6:ω-3 PUFA ratio of 15:1, far from the ideal ratio of 4:1 [209]. It is well-established that high consumption of LA (ω-6)-enriched foods drives an abnormal eicosanoid biosynthesis, involved in the etiology of many chronic diseases, while a diet rich in ω-3 PUFAs has a protective role against cardiovascular and NDDs [210,211,212]. Interestingly, the adherence to MedD permits a congruous intake of ω-3 PUFAs, widely distributed in plant-derived foods including walnuts, purslane, legumes, green leafy vegetables, and plant oils such as olive oil [213,214]. Animal-derived ω-3 is also important: in the ecosystem, algae are the primary producers of DHA and EPA, which, once they have entered the food chain through marine phytoplankton, accumulate in fish, especially in fish oil (salmon, trout, mackerel, herring and sardines) [215]. ECs seem to be involved in the neuroprotective properties of dietary ω-6:ω-3 ratio [205]. For example, a chronic consumption of ARA (ω-6)-enriched foods is associated with an increase in EC levels, causing desensitization and downregulation of CBRs [5]. On the other hand, a dietary supplementation of ω-3 PUFAs enhances plasma and tissue concentrations of EPA and DHA, balancing AEA and 2-AG amounts [216]. In light of these observations, one of the main problems related to chronic exposure to a typical Western diet may be a marked uncoupling between the expression and localization of CBRs and the uncoordinated synthesis of endogenous ligands. Diets with high amount of ω-3 PUFAs have been shown to increase the expression of CBRs and biosynthetic enzymes, such as NAPE-PLD, DAGLα and DAGβ, in mice [217]. ECs are key players in synaptic plasticity regulation [218]. In mice, dietary deprivation of ω-3 PUFAs leads to anxiety-like behavior and disrupts the CB1R signaling pathway (ERK1/2) in the prefrontal cortex, hypothalamus and hippocampus [219]. In addition, chronic ω-3 deficiency interferes with endocannabinoid-mediated long-term synaptic depression (LTD) in the prefrontal cortex and nucleus accumbens, and inhibits hippocampal endocannabinoid-mediated inhibitory LTD [220,221]. DHA (ω-3) treatment, in cultured hippocampal neurons, promotes TRPV1 and CB1R expression, while in vivo it improves spatial memory in rats (150 or 300 mg DHA/kg/day) [222]. Taken together, these findings could explain some of the benefits observed in patients with mild cognitive impairment (MCI) who implemented ω-3 PUFA consumption [223].

Furthermore, the endocannabinoid system could undergo a progressive age-related deficit, accelerating cognitive decline [224]. These hallmarks of brain aging could be attenuated by ω-3 PUFA supplementation [222,225]. The stimulation of neurogenesis is an important therapeutic target in NDD treatment: EPA seems to stimulate neural stem cell proliferation via the CBRs/p38 MAPK signaling cascade, while DHA could be involved in neural stem cell differentiation [226]. Recent studies have focused on a series of ω-3 endocannabinoids (ω-3ECs), such as DHEA and EPEA [159], produced by DHA and EPA conversion through the same AEA biosynthetic pathways [227]. DHEA and EPEA act as partial agonists of CBRs and PPAR-γ [228]. These ω-3ECs properties could induce the “entourage effect”, boosting AEA and 2-AG biological activity through their partial affinity for the same receptors and enzymes [205]. DHEA, named “synaptamide”, can also bind G protein-coupled orphan receptor (GPR110), promoting neurogenesis, neuronal differentiation and synaptogenesis via cAMP-dependent pathways [229]. Moreover, DHEA has been shown to reduce neuroinflammation, both in vitro and in vivo, enhancing cAMP/PKA signaling and inhibiting NF-κB activation [230].

## 6. Between “Entourage” Effect and Noncanonical CBRs: *N*-Acylethanolamines (NAEs) and Resveratrol

*N*-acylethanolamines (NAEs) are a series of lipid mediators, belonging to the same class as AEA, derived from the common precursor *N*-acylated phosphatidylethanolamine (NAPE) [45]. NAEs are pleiotropic compounds produced on demand both in the CNS, in which their levels greatly exceed those of AEA, and in peripheral tissues [231,232]. This “extended” EC family, including PEA, OEA and stearoylethanolamide (SEA), shares with AEA the same biosynthetic and catabolic enzymes. Specifically, NAEs can be the substrate of FAAH-1 and FAAH-2 isoforms [233,234]: FAAH-deficient mice have increased tissue levels of both AEA and NAEs [235]. NAAA is a second enzyme located in lysosomes, deputed to NAE degradation at low pH, although a preference for PEA was demonstrated [236,237]. In addition to canonical CBRs, other receptors have been proposed as targets of ECs, and for some of them, NAEs would represent potential ligands. For example, OEA and PEA activate TRPV1, while PEA could act as an orphan GPCR 55 (GPR55) agonist [238,239]. Despite these similarities with AEA, these endocannabinoid-related mediators lack affinity for CBRs [240]. The concentrations of NAEs, as well as NAPEs, are typically increased during brain injury, suggesting a possible neuroprotective role [241]. Two possible mechanisms are the “entourage effect” and the direct receptor activation. The first one is mediated by AEA accumulation following FAAH-competitive inhibition and/or FAAH downregulation [242,243]. The second one involves NAEs binding to specific receptors: PEA acts as a GPR55 agonist, which has been linked to several downstream signaling events, including CREB, NF-κB and ERK1/2 phosphorylation [244]; TRPV1, activated by both PEA and OEA, modulates neuroinflammatory processes [245]; and NAEs enhance PPAR-α signaling, promoting anti-inflammatory response and neuroprotection [246,247]. According to in vitro and in vivo studies, increasing NAE levels may represent a viable therapeutic option for NND treatment [240]. For example, OEA and PEA have been shown to reduce ROS production, preventing mitochondrial OxS [248,249]. Since its discovery, PEA has been considered an anti-inflammatory agent: it promotes microglia motility, probably in a PPAR-α-mediated manner [250]. Moreover, its analgesic and anti-inflammatory properties seem to require CBRs and TRPV1, through the “entourage effect” [232]. In both in vitro and in vivo models of PD, OEA exerts neuroprotection against 6-OHDA-induced degeneration of substantia nigra [251]. OEA and PEA prevent LPS-induced NF-κB activation, iNOS and COX-2 expression, and OxS in rats’ frontal cortex [252]. NAEs can also lead to pro-resolving signal synthesis, such as IL-10 and IL-1 receptor antagonists [253]. Despite NAE levels being elevated during brain injuries, they may not be enough for an adequate response [254]. Growing evidence suggests that an exogenous administration of NAPEs and NEAs may have a clinical relevance for many NDDs [255]. In EAE mice, exogenous administration of PEA reduces inflammation, demyelination and neuronal damage with a general improvement in clinical phenotype [256]. In both AD and PD in vivo models, PEA supplementation mitigates inflammatory response, OxS and neuronal apoptosis, ameliorating behavioral impairment [257,258]. In a similar fashion, OEA supplementation showed efficacy in counteracting neurobehavioral changes in stress-related disorders as well as in chronic neuroinflammatory diseases [259,260]. In light of these observations, encouraging intake of NAEs and/or their precursor NAPEs can boost their tissue levels, which in turn can affect the progression of many NDDs through modulation of the endocannabinoid system and PPAR-α [255]. It is known that high-fat diets (e.g., Western dietary style) reduce intestinal and hippocampal concentrations of ECs and NAEs [261,262]. On the other hand, strict adherence to MedD induces EC and NAE synthesis. Artmann et al. (2008) investigated five different types of dietary fats, evaluating their short-term effects on the endocannabinoid system: the diet rich in olive oil (MedD-like) increased brain levels of OEA, LEA and AEA in rats [208]. In a recent study, dealing with a cross-sectional sample of 195 healthy men and women, the levels of several lipid mediators belonging to the endocannabinoid system were assessed before and after two days of MedD. A short-term change in eating style and fatty acid intake was found to increase circulating levels of NAEs [263]. Moreover, the EC, NAE and NAPE contents of 43 food products were evaluated in order to simulate their daily intake in three different diets: MedD had the highest amount of NAPEs and NAEs compared with a Western diet (263 vs. 163 mg/day and 0.25 vs. 0.08 mg/day, respectively), with the same EC content (0.17 mg/day), which was higher than a vegetarian diet (0.01 mg/day) [264]. Plant-food products are rich in NAPEs and NAEs, especially refined wheat flour, which seems to be the main source of NEAs, followed by legumes (beans, lentils and chickpeas). Extra-virgin olive oil contains approximately 53.9 ng/g dw of LEAs and 85.7 ng/g dw of OEAs. On the other hand, animal food products are the primary source of dietary ECs [264]. RSV (3,4’,5 trihydroxy-trans-stilbene) is a natural polyphenol, contained in several plant-derived foodstuffs of MedD, such as grapes, peanuts and berries (blackberries, blackcurrants, blueberries, cranberries) [265]. Since red grape skin contains the highest concentration of RSV, red wine represents its most concentrated dietary source (mean RSV content of 6 mg/L) [266]. In recent decades, it has proved to be a promising therapeutic strategy against cardiovascular diseases and NDDs [267,268]. However, the protective effects of red wine follow a U-curve: the greatest benefits seem to be associated with a low-to-moderate consumption (10–50 g/day for men, 5–25 g/day for women), within a Mediterranean pattern [269]. A critical therapeutic limitation is RSV’s poor bioavailability, due to its rapid conversion into glucuronated and sulfonated conjugated forms [266]. On the other hand, it has some attractive characteristics that make it particularly interesting for NDD treatment: RSV can cross the BBB and, binding Sirt1, enhance its activity by about two times [270,271]. As previously reported, Sirt1 plays important roles in neuropathology, improving neuronal antioxidant reserves as well as promoting an M2 microglial shift [108]. According to epidemiologic evidence, modest wine consumption protects against AD onset [272]. Likewise, works on animal models of HD, PD and amyotrophic lateral sclerosis (ALS) have confirmed the neuroprotective properties of RSV [273]. Beyond classic immunomodulatory and antioxidant action, part of RSV’s neurotrophic effects may be mediated by the endocannabinoid system.

The finding that arachidin-1, arachidin-3 and piceatannol (RSV analogs) bind human and mice CBRs was the first evidence to suggest the abovementioned hypothesis [274]. Hassanzadeh et al. (2016) investigated RSV’s antidepressant activity in male Wistar rats, comparing its neurotrophic effects with amitriptyline and clonazepam: after four weeks of treatment, RSV enhanced nerve growth factor (NGF), 2-AG and AEA release at the level of the prefrontal cortex, hippocampus, amygdala and olfactory system. Interestingly, pre-treatment with a CB1R antagonist (AM251) prevented an increase in NGF levels [275]. NGF is a neurotrophin, produced in the cortex and hippocampus and involved in cholinergic–hippocampal interactions, as well as in the synaptic plasticity and neurogenesis of the hippocampus [276,277]. Low NGF levels are associated with AD progression, while their elevation has been shown to improve cognitive performance [278]. In a murine model of neuropathic pain, RSV exerted antinociceptive activity through AEA and 2-AG, in a CB1R- and μ-opioid receptor-mediated manner [279]. In another study, the neuroprotective properties of a single acute dose of RSV were tested against bilateral common carotid artery occlusion, followed by reperfusion challenge. In the frontal cortex, RSV pre-treatment (i) controls lipoperoxide levels induced by hypoperfusion/reperfusion stress; (ii) increases DHA and PEA levels; (iii) triggers CBR and PPAR-α expression; and (iv) drives the synthesis of syntaxin-3 and post-synaptic density protein-95 (PSD-95), two proteins involved in synaptic plasticity [280] (Table 1).

## 7. EC Activity and Diseases

Recently, much interest has been focused on the role of endocannabinoids in peripheral tissues, beyond the central and peripheral nervous system, as an increasing number of functions have been attributed to this class of compounds, and they are now considered a potential candidate for the treatment of several diseases.

## 8. EC Induction of Obesity Counteracted by MedD-Related Compounds

One important issue has been the establishment of ECs’ orexigenic function, mediated by binding to CB1Rs distributed within the hypothalamus, as evidenced by an increase in food intake following AEA administration [287]. EC levels raise during starvation while, when satiety is reached, their concentration in the hypothalamus and the mesolimbic areas slowly decreases [287]. High-calorie dietary intake induces endocannabinoid hypertone and CB1R hyperactivation. In previous in vivo studies with obese mice and clinical trials in overweight individuals, treatments with CB1R antagonists or reverse agonists exerted an antiobesity effect, improving metabolism-related risk factors [6,288]. In obese mice treated with a CB1R antagonist, a significant reduction in food intake and a broad improvement in metabolic parameters were found [289,290]. Increased levels of AEA in the plasma of women with binge eating syndrome confirmed the endocannabinoid system hyperactivity related to an excessive food intake [291]. In the past, this endocannabinoid action conferred an evolutionary benefit because hyperphagia ensured an accumulation of energy within adipose tissue, which was essential for survival during periods of famine. High-sugar diets, promoted by the advancement of agricultural practices, have disrupted the homeostatic role of CB1Rs [160,292,293], but CB2Rs may have a protective role, counteracting the metabolic impact of CB1R dysregulation [160]. In modern societies where the Western diet is widespread, an increased food intake results in the accumulation of visceral fat, leading to the development of obesity, metabolic syndrome and type 2 diabetes, the main risk factors for the development of cardiovascular diseases [6,160,287,294].

In obesity, the imbalance of endocannabinoid peripheral signaling also depends on a reduction in FAAH metabolites, which impacts visceral fat, leading to CB1R overstimulation and subsequent adiponectin inhibition, resulting in the exacerbation of insulin resistance and low-grade inflammation [288,295]. CB1R overactivity associated with hyperphagia was found in obese patients with Prader–Willi syndrome and in obese mouse models, and peripheral CB1R antagonist treatment promoted weight loss [296]. A good balance of omega-6 (ω-6) and ω-3, needed for a healthy diet, is not found in Western diets [5]. Important PUFAs, such as LA, which are then converted to arachidonic acid, increase levels of ECs chronically, inducing a subsequent downregulation and desensitization of their receptor system, contributing to obesity, as shown in mouse models [297]. In obese mice supplied with a diet rich in arachidonic acid, the introduction of ω-3 fatty acids lowered endocannabinoid tone in adipose tissues and liver. Thus, it appears that dietary intake of ω-3 restores endocannabinoid signaling and also “entourage” compounds such as PEA and OEA [5]. In a 2012 study, mice on an LA diet showed adipogenesis and EC overactivity; supplementation with EPA and DHA restored the physiological levels of AEA and 2-AG [297].

Common probiotics, such as Lactobacillus acidophilus and Bifidobacterium, which are found in foods including yogurt, increase CB2R expression [298] and downregulate CB1R expression by reducing their mRNA levels, as shown in obese mice [299].

## 9. ECs and Gut Microbiome

A correlation between ECs and microbiome has recently been identified. Changes in eating habits can induce the development of metabolic disorders by influencing the crosstalk between ECs and the gut microbiota [300].

Obesity promotes changes in the gut microbiota, with a reduction in the Bacteroidetes/Firmicutes ratio followed by enteric mucus degradation, thinning of the intestinal barrier, and increased gut permeability, ending with the development of low-grade chronic inflammation of the intestinal mucosa. LPS released by bacteria increases AEA levels, targeting FAAH expression [301].

A CB_1_ receptor blockade restored the gut barrier in obese mice by improving the expression of tight junction proteins [299]. A cannabinoid hypertonus would therefore lead to an impairment of gut permeability with an increased circulation of bacterial endotoxins, and a consequent stimulus in the synthesis and release of pro-inflammatory cytokines. A further consequence may consist in the increased transfer of these cytokines to the brain, by an increase in BBB permeability or even by transport through the afferent fibers of the vagus nerve (gut–brain axis) [302,303]. In fact, rimonabant, which antagonizes CB1R, reduces LPS levels and inflammation [304].

## 10. ECs and Inflammation

One of the main roles of ECs concerns the modulation of inflammatory processes and cells belonging to the immune system, therefore affecting all diseases based on chronic inflammation including cancer, diabetes mellitus and atherosclerosis, as well as cardiovascular, chronic airway, inflammatory bowel, autoimmune and neurodegenerative diseases [305].

Increasing the levels of AEA and 2-AG in the brain with the use of inhibitors of the main degrading enzymes, fatty acid amide hydrolase (FAAH) and monoacylglycerol lipase (MAGL), has been documented as an effective strategy to control the immune response in different models of MS, Huntington’s disease (HD) and AD.

## 11. ECs and Cancer

Anandamide and 2-AG exert a pro-apoptotic effect in cultured tumor cells and inhibit their migration, holding the potential to modulate tumor growth [306]. Several studies have shown that ECs as well as phytocannabinoids and synthetic cannabinoids exert an inhibitory effect on cancer growth by inducing apoptosis or cell cycle arrest, or by inducing autophagic cell death by inhibition of mTORC1. Furthermore, several inhibitors of EC cathabolic enzymes suppressed cancer aggressiveness [307]

## 12. Endocannabinoid Tone Modulation by Plant Flavonoids

Plant flavonoids, including BCA and Kmp, modulate endocannabinoid tone through FAAH inhibition; show antioxidant, antidiabetic and anti-inflammatory activity; and have been suggested to decrease risk factors for cardiovascular disease, although this role is still under debate [176,195]. RSV interacts with both CB1Rs and CB2Rs [171]. RSV activates SIRT1 and was shown to have an antiobesity effect in mice and rats through inhibition of PPAR-γ in adipocytes [308,309,310,311,312]. RSV modulates glucose uptake in T2DM via AMPK activation [309,313]. Even though in vitro and in vivo studies seem to demonstrate the therapeutic efficacy of RSV, clinical studies do not always confirm these results in both obese and diabetic subjects [308,314]. Many studies have shown the anti-inflammatory properties of RSV [315]: in rats with an inflamed colon, RSV suppressed production of the pro-inflammatory cytokines COX-1 and COX-2 [316], and in murine and human inflamed adipose tissue, RSV was found to lower the expression of TNFα, IL-1β, IL-6 and IL-8 [317,318]. In addition, RSV obtained from a grape extract decreased levels of pro-inflammatory cytokines in peripheral blood mononuclear cells after the intake of a high-calorie meal in healthy subjects as well as in diabetic and hypertensive subjects [319,320]. In a rat model of nonalcoholic steatohepatitis induced by a high-fat diet, RSV intake inhibited CB1Rs in the colon and diminished intestinal inflammation through activation of CB2Rs, suggesting its role in preserving intestinal barrier integrity by modulation of ECs [321].

Interestingly, there are relatively few plant-based secondary metabolites that trigger CB1Rs compared with those that activate CB2Rs [283]. Overall, CB2R activation by cannabimimetics in plant-based diets has a salutary effect against obesity-induced risks, as they counteract the CB1R effects. [160,322].

ECs, via PPAR-γ, physiologically control adipocyte maturation and lipid storage, exerting a positive effect against lipotoxicity, which induces insulin resistance, risk of diabetes, and cardiovascular and hepatic pathologies related to metabolic syndrome [323]. Vegetable-based diets seem to have a protective role, triggering PPAR-γ activation [324].

BCP induced PPAR-γ-mediated anti-inflammatory effects in mice with colitis triggered by sodium dextran sulfate (DSS) [325]. In LPS-stimulated human monocytes, BCP caused a significant reduction in ERK1/2 and JNK1/2 and a reduction in the serum levels of TNF-α and IL-1β. In CB2R knockout mice, BCP treatment did not decrease inflammation caused by carrageenan, indicating the importance of CB2R signaling [92]. In rats with diabetes mellitus, high doses of BCP decreased inflammation and OxS induced by hyperglycemia [326,327]. BCP selectively binds peripheral CB2Rs with a cannabimimetic activity, with a positive activity on inflammatory processes [328,329].

In animal models of colorectal carcinogenesis, a diet based on olive oil, ω-3-/ω-6 fatty acids, showed antiobesity and anti-inflammatory activity through the interaction with CB2Rs, helping adipogenic homeostasis and decreasing endocannabinoid activation [330,331].

Further insights into the molecular processes described above (see Table 1) may result in new therapeutic strategies. In light of these observations, a healthy lifestyle associated with the MedD seems to be a key factor in the modulation of the endocannabinoid system, which counteracts inflammation and impacts CNS functions [12,332].

## 13. Conclusions

The endocannabinoid system is a pleiotropic complex of endogenous pro-homeostatic mediators [333], and its suboptimal functioning is associated with many diseases, including NDDs [5]. “Clinical endocannabinoid deficiency syndrome” was the definition that was used to describe these pathological conditions [334]. From a phylogenetic perspective, the co-evolution of dietary habits and human physiology could explain how many phytochemicals and MedD foodstuffs modulate the endocannabinoid system: McPartland et al. (2007) suggest that diet represents a key player in the shaping of certain endocannabinoid genes [335]. The mismatch between a high-calorie diet, such as the Western diet, and ancient genes, adaptive among our ancestors during food restriction and hunting, leads to chronic metabolic disorders and NDDs [160]. CB1R overstimulation, abnormal activation of the endocannabinoid system and its subsequent downregulation may be also consequences of an excessive consumption of foods rich in refined carbohydrates and fats [5,264]. On the other hand, a high consumption of vegetable foodstuffs and spices, typical of the MedD, contains the appropriate amounts of CB2R agonists and cannabimimetic components necessary to counter CB1R-mediated metabolic stress [160]. Additionally, it may restore the uncoupling between the expression and localization of CBRs as well as reducing neuroinflammation, OxS and neuroapoptosis, via multiple downstream cascades in the CNS (Table 2). Despite the small number of studies and the few human trials conducted to evaluate the in vivo endocannabinoid-mediated neuroprotection provided by some cannabimimetic compounds of the MedD, preclinical studies have shown that they may represent a promising therapeutic strategy to slow down the progression of many NDDs. Their metabolism and bioavailability are also critical aspects to consider in any dietary intervention.

To our knowledge, this is the first review that aims to evaluate the evidence accumulated in the literature on the neuroprotective, immunomodulatory and antioxidant properties of the Mediterranean lifestyle related to the modulation of the endocannabinoid system. In light of this nutraceutical paradigm, a new prospect for research and clinical interventions against neurodegenerative diseases based on a specific dietary protocol is suggested.

## Figures and Tables

**Figure 1 biomolecules-11-00790-f001:**
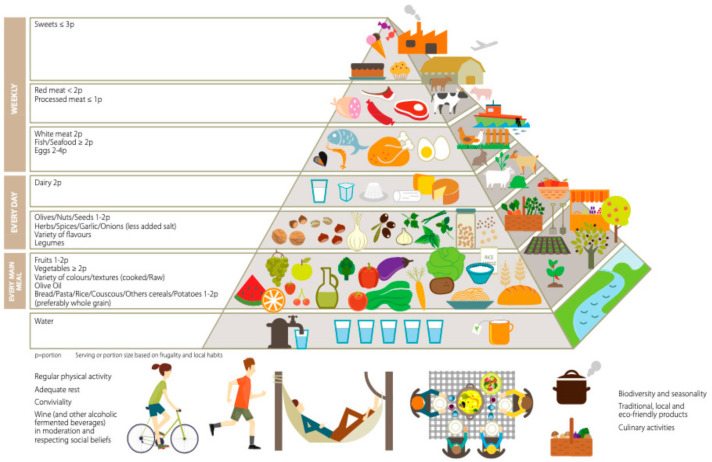
New Pyramid for a Sustainable Mediterranean Diet.

**Figure 2 biomolecules-11-00790-f002:**
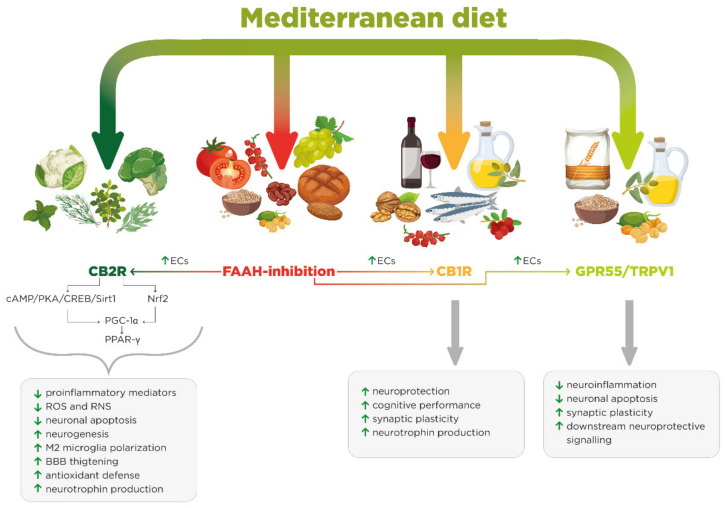
Mediterranean diet activity on EC receptors (CB1R, CB2R, GPR55/TRPV1) and on FAAH inhibition.

**Figure 3 biomolecules-11-00790-f003:**
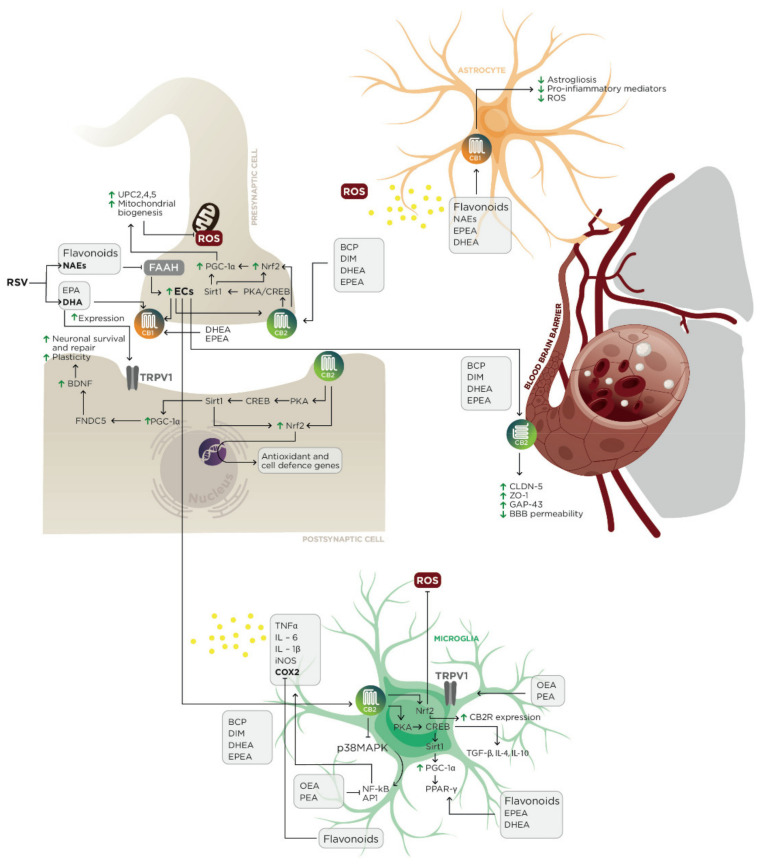
Activity of Mediterranean diet components on the endocannabinoid system at the level of CNS cells.

**Table 1 biomolecules-11-00790-t001:** The potential role of Mediterranean diet components in the modulation of the endocannabinoid system.

Class of Molecules	Compound	Dietary Origin	Endocannabinoid Target/Effects	Potency	Proposed Oral Dose in Humans
Bicyclic sesquiterpene	β-caryophyllene (BCP)	Black pepper, hops, oregano, cloves, valerian, wild sage, rosemary, basil and cow’s milk	CB2R full agonist	100 nM [196]	4 mg/kg/day [196]
Glucosinolates (GLs)	3,3′-diindolylmethane (DIM)	Broccoli, cabbage, cauliflower and horseradish	CB2R partial agonist	1 μM (binding affinity) [161]	0.8–1.6 mg/K [149,150]
Flavonoids	Biochanin A (isoflavone)	Peanuts and chickpeas	FAAH inhibitor	2.4 μM (IC50) [176]	50 mg/kg (rats) [281]
	Genistein (isoflavone)	Whole grain bread, beans, lentils, chickpeas, split chicklings and fava beans	FAAH competitive inhibitor/PPAR-γ agonist	2.8 μM (Ki value) [182]	9.0 mg/day [189]( based on an Asian population)
	Daidzein (isoflavone)		FAAH competitive inhibitor/PPAR-γ agonist	1.7 μM (Ki value) [182]	7.4 mg/day [189] (based on an Asian population)
	Apigenin (flavons)	Parsley, onions, celery, chamomile, oregano, thyme, basil, oranges, tea, beer and wine	FAAH competitive inhibitor/PPAR-γ agonist/COX-2 inhibitor	15 μM (IC50) [166]	0.7 mg/kg [191]
	Kaempferol (flavonol)	Leafy vegetables, apples, onions, broccoli, berries, tea, cabbage, broccoli, endive, kale, beans, tomato, strawberries, leek and grapes	Competitive FAAH inhibitor/PPAR-γ agonist	5 μM (Ki value) [166]	1.4–5.7 mg/kg/day [195]
ω-3 PUFAs	DHA/EPA	Walnuts, purslane, legumes, green leafy vegetables, olive oil, salmon, trout, mackerel, herring and sardines	↑ CBRs NAPE-PLD, DAGLα and DAGβ expression (DHA/EPA); CBRs/PPAR-γ partial agonists (DHEA/EPEA); GPR110 agonist (DHEA)	DHEA → CB_1_ 1044 nM (EC50), CB2 305 nM (EC50); EPEA → CB_1_ 0.1 nM (EC50), CB2 2.1 nM (EC50) [282]	>1 g/day DHA/EPA [283,284]
*N*-acylethanolamines (NAEs)	Palmitoylethanolamine (PEA)	Refined wheat flour, beans, lentils, chickpeas and extra-virgin olive oil	Competitive FAAH/NAAA inhibitor, TRPV1/PPAR- α/GPR55 agonist	/	600–1200 mg/day [285]
	Oleoylethanolamide (OEA)		Competitive FAAH/NAAA inhibitor, TRPV1/PPAR-α agonist	/	125 mg/day [286]
Polyphenols	Resveratrol (stilbene)	Red grapes, red wine, peanuts, blackberries, blackcurrants, blueberries and cranberries	↑ DHA and PEA levels; ↑ CBRs and PPAR-α expression; CB1R-mediated neuroprotection	/	1.4–4.2 mg/kg/day [195]

**Table 2 biomolecules-11-00790-t002:** A synoptic overview of neuroprotective actions, mediated by endocannabinoid system modulation, by the Mediterranean diet in in vivo models of neurodegenerative disorders.

Disorders	Disease Models	Compound	Effects	
AD	Transgenic APP/PS1 mice	BCP	↑ Cognitive performance ↓ Aβ accumulation ↓ Astrogliosis ↓Pro-inflammatory mediators	[96]
AD	ICR mice (scopolamine)	DIM	↑ BDNF ↑ Antioxidant enzyme expression ↑ Cognitive performance ↓ Neuroapoptosis	[150]
AD	Transgenic APP/PS1 mice	Apigenin	↑ Neuroprotection	[191]
AD	Tg 2576 mice	RSV	↑ Cognitive performance ↓ Amyloid brain pathology	[273]
AD	Mouse model p25	RSV	↓ Hippocampal neurodegeneration ↑ Cognitive performance, ↑ SIRT1 overexpression	[273]
AD	TgSwDI mice	EVOO (OEA)	↓ Aβ and neurofibrillary tangles ↑ Cognitive performance BBB Aβ clearance	[76]
AD	C57BL6 mice (Aβ 25–35 intracerebroventricular injection)	PEA	↓ Neuroapoptosis ↓ Neuroinflammation ↓ Memory dysfunction ↓ OxS	[257]
PD	Wistar rats (rotenone-induced toxicity)	BCP	↓ Neuroapoptosis ↓ iNOS), COX-2, TNFα, IL-6 and IL- 1β expression ↑ Antioxidant defenses	[97]
PD	C57BL/6J mice (MPTP)	DIM	↓ Neuroapoptosis ↑ Cytoprotection	[156]
PD	Wistar rats (6-OHDA)	OEA	↑ Neuroprotection	[251]
PD	PPAR-αKO and PPAR-αWT mice (MPTP)	PEA	↓ Microglial activation ↓ Microtube alteration ↓ Neuroapoptosis ↑ Motor behavior	[258]
MS	EAE mouse model	BCP	↓ Axonal demyelination ↓ Microglial cells, CD4+ and CD8+ activity ↑ Th1/Treg immune balance	[98]
MS	EAE mouse	PEA	↓ Demyelination ↓ Neuronal damage ↑ Clinical phenotype	[256]
MS	EAE mouse model	BCP	↓ NO and H_2_O_2_ production	[112]
ALS	Mouse model of ALS	RSV	↑ Neuroprotection	[273]
Neuropsychiatric diseases	C57BL6/J mice (ω-3 deficiency)	ω-3	↑ Anxiolytic effect	[219]
Obesity	ω-3 deficiency	ω-3	↑ CBR-Gi/o coupling↑ Mood modulation	[220]
Obesity	Obese mice	ω-3	↑ AEA/2-AG balance	[5]
Memory impairment	P18–P24 CD1 mice (ω-3 deficiency)	ω-3	↑ Hippocampal LTP ↑ Synaptic plasticity ↓ Hippocampal LTP	[221]
Brain aging	Old Wistar rats (EPA/DHA supplementation)	EPA/DHA	↓ Neuroapoptosis ↑ Neuroprotection ↑ PPAR-γ expression	[225]
LPS-induced neuroinflammation	C57BL/6J mice	DHEA	↑ cAMP/PKA signaling ↑ Inhibition of NF-κB activation ↓ Pro-inflammatory mediators	[230]
LPS-induced neuroinflammation	Wistar Hannover rats	OEA/PEA	↓ Oxs ↑ NF-κB inhibition ↓ Pro-inflammatory mediators	[252]
LPS-induced neuroinflammation	C57BL/6 mice	DIM	↓ iNOS and COX-2 expression ↑ NF-κB inhibition	[152]
Neuropathic pain	C57BL/6 mice (formalin)	BCA	↓ ERK phosphorylation ↓ Pain	[176]
Neuropathic pain	Swiss mice	RSV	↓ Pain ↑ Neuroprotection	[279]
Brain atrophy	Wistar rats	RSV	↑ NGF	[275]
Occlusion-reperfusion stress	Wistar rats	RSV	↓ OxS ↓ Neuroapoptosis ↑ DHA and PEA levels ↑ CBR and PPAR-α expression ↑ Synaptic plasticity	[280]
Stress-related disorders	Wistar rats (contextual fear conditioning)	Kmp	↑ Anxiolytic effect	[201]

## Data Availability

Not applicable.

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
