# Peer review of "Mediterranean Diet and Neurodegenerative Diseases: The Neglected Role of Nutrition in the Modulation of the Endocannabinoid System"

_biomolecules, 2021, doi:10.3390/biom11060790_

Round 1

Reviewer 1 Report

The article is devoted to the actual problem of the influence of the Mediterranean diet on the state of the endocanabioid system. In general, the work makes a good impression and is of high interest to the reader. The review article presents numerous modern data focusing on various aspects of the Mediterranean diet as the main source of numerous elements of plant and animal origin, their biosynthesis in the body and the main metabolic pathways, as well as the effect of these components on the pathophysiology of neurodegenerative diseases.
Attention is drawn to a significant size of the review article and a large amount of analyzed literature, as well as a complex multi-level understanding of the problem under study. The complex schemes presented in the review article deserve high marks.
Comments that arose on the text of the manuscript:
In the section “Endocannabinoid system: physiology and pathophysiology”, the authors should not only present the factual data of the literature on the issue, but also more deeply analyze the information presented, as well as summarize them, formulating the main problem areas in studies of the endocanabinoid system.

The Mediterranean diet section on page 6, pp. 229-250 is for some reason presented in Italian. It is necessary, if the authors have no serious reasons for this, to submit this fragment of the text in English. However, the content of this fragment looks somewhat random. Needs to be edited.
The entire Mediterranean diet section needs to be edited and stylistically revised.
In my opinion, the section "Endocannabinoids and diseases" needs more generalization of the presented literature data and supplementing the conclusions made by the authors of such a generalization at the end of the section.

Author Response

Please see the attachment "Reply to reviewer 1"

Reviewer 2 Report

This is a well referenced report. It contains a large amount of information.

The graphs and table are well described.

 On page 6 by error there is a whole page in Italian?- describing it as scope of revision. 

A better organization could be suggested to move from in vitro, to in vivo models. This is should be separated to AD, MS, PD with respect to the data  generated. Finally, human studies when they are available.

Addition of tables of disease models and CBD like molecules intervention would clarify potential clinical application. Specifically, in these models the diet is otherwise fixed in contrast to the innate variety of human diet. 

Can it be better specified that there is a specific diet/CBD like supplement for a given neurological disease or it is highly similar for exerting reparative or preventive role on all given disease? Or alternatively changing a proportion of a MedD diet for a given disease.   

It is clear that diet is a highly complex interaction of ingredients and therefore specifically defining which ingredient caused the improvement in outcome is not possible.

The diet pyramid advocates whole wheat or whole rice, and peanuts- however there is evidence those produce lectin which have proinflammatory effect causing leaking gut that can affect the brain as well. This is not mentioned in the paper. 

Author Response

Please see the attachment: Reply to Reviewer 2"

Round 2

Reviewer 1 Report

The authors of the review article were attentive to the comments made and significantly improved the content of the manuscript. Most of the comments were taken into account and corrected in the new version of the review article. The edited version after the stylistic revision of the English language of the work can be recommended for publication in Biomolecules.

Reviewer 2 Report

The paper addressed the comments. Diet is complex and over exposure to one type is not beneficial. The totality counts. 

Also there is no recommendation to treat a specific neurologic disease.